# A redox-neutral catechol synthesis

Qian Wu[1,*], Dingyuan Yan[2,*], Ying Chen[1,*], Ting Wang[3], Feng Xiong[2], Wei Wei[2], Yi Lu[2], Wei-Yin Sun[2], Jie Jack Li[4] & Jing Zhao[1,2]

Ubiquitous tyrosinase catalyses the aerobic oxidation of phenols to catechols through the binuclear copper centres. Here, inspired by the Fischer indole synthesis, we report an iridium-catalysed tyrosinase-like approach to catechols, employing an oxyacetamide-directed C–H hydroxylation on phenols. This method achieves one-step, redox-neutral synthesis of catechols with diverse substituent groups under mild conditions. Mechanistic studies confirm that the directing group (DG) oxyacetamide acts as the oxygen source. This strategy has been applied to the synthesis of different important catechols with fluorescent property and bioactivity from the corresponding phenols. Finally, our method also provides a convenient route to $^{18}$O-labelled catechols using $^{18}$O-labelled acetic acid.

[1] Guangdong Key Lab of Nano-Micro Material Research, School of Chemical Biology and Biotechnology, Peking University Shenzhen Graduate School, Shenzhen 518055, China. [2] State Key Laboratory of Coordination Chemistry, Department of Chemistry, Institute of Chemistry and BioMedical Sciences, School of Chemistry and Chemical Engineering, Collaborative Innovation Centre of Chemistry for Life Sciences, Nanjing University, 163 Xianlin Avenue, Nanjing 210023, China. [3] School of Chemical Biology and Biotechnology, Peking University Shenzhen Graduate School, Shenzhen 518055, China. [4] Department of Chemistry, University of San Francisco, 2130 Fulton Street, San Francisco, California 94117, USA. * These authors contributed equally to this work. Correspondence and requests for materials should be addressed to J.Z. (email: jingzhao@nju.edu.cn).

Catechols are essential bioactive molecules in human metabolism and normal physiological activities, acting as effective structural units in many bronchodilator, adrenergic, anti-parkinsonian and anti-hypertensive drugs (Fig. 1a)[1]. For example, L-dopa is a well-known drug used for the treatment of Parkinson's disease and is converted from L-tyrosine via catalysis by tyrosine hydroxylase in the central nervous system under mild conditions. The dopamine system plays a central role in several significant medical conditions, including attention deficit hyperactivity disorder (ADHD), schizophrenia and addictions[2]. On the other hand, estradiol is important in the regulation of the female oestrous and menstrual reproductive cycles[3].

Synthesis of catechols usually requires lengthy steps and harsh reaction conditions, with low selectivity and conversion rates[4]. To develop efficient routes to synthesize catechols, we aim to construct a new C–O bond at the *ortho* position of phenols through directed C–H bond oxygenation. The catalytic transformation of the C–O bond from benzene to phenol is considered to be one of the greatest challenges in catalytic chemistry[5]. Tremendous progress has been achieved in directed C–H hydroxylation/oxygenation by metal catalysis with $O_2$ (refs 6–9), $H_2O$ (refs 10–12), peroxides[13–18] or the *in situ* hydrolysis of newly installed acyloxy groups as oxygen sources such as OAc, OTFA and so on refs 19–30 (Fig. 1b). Notably, Yu and colleagues[6] first developed a versatile Pd-catalysed *ortho*-hydroxylation of benzoic acids at 1 atm of $O_2$ or air under non-acidic conditions. Gevorgyan and colleagues[31] recently reported an ingenious silanol-directed Pd-catalysed C–H oxygenation of phenols followed by desilylation of the silacyle with tetrabutylammonium fluoride (TBAF), furnishing catechols with $PhI(OAc)_2$ as the oxygen source. Rao and colleagues[32]

demonstrated the efficient ruthenium(II)- and palladium(II)-catalysed C–H hydroxylation of aryl carbamates and a subsequent deprotection to afford catechols in good yields. These pioneering works used TFA/TFAA as the oxygen source[26].

We have recently reported a number of metal-catalysed C–H bond functionalizations based on a powerful directing group, -O–NHAc (oxyacetamide), which can act as an internal oxidant to yield versatile phenol derivatives[33,34]. A variety of substituted substrates with -O–NHAc directing group were easily prepared from phenols or aryl boronic acid compounds[35–38]. Inspired by the classic Fischer indole synthesis[39–41], we hypothesized that catechols would be furnished from *N*-phenoxyacetamides provided that the acetamide group could function as an intramolecular oxygen source. By choosing appropriate metal catalysts, the acetamide group could function as a directing group, an internal oxidant and an oxygen source simultaneously (Fig. 1c)[38,42–45]. This design would avoid the need for an external oxidant, which could be detrimental to the fragile polyphenols. Notably, seminar reports from the Cheung and Buchwald[46], and Zhao and colleagues[47] clearly established that the acetamide group could participate in C–O bond formation.

Herein, we report an iridium-catalysed, one-step, mild and redox-neutral synthesis of catechols from acetamide-protected phenols, employing a bioinspired intramolecular oxygen transfer strategy.

## Results

**Model studies**. We first explored Zn-, Al-, Fe-, Cu-, Pd- and Ru-catalysed systems to *N*-phenoxyacetamides (**1a**). Unfortunately, the target product (**2a**) was not detected. The same result was shown in the common Rh-catalysed system for -O–NHAc (Table 1, entries 1–10 and Supplementary Table 1). Considering

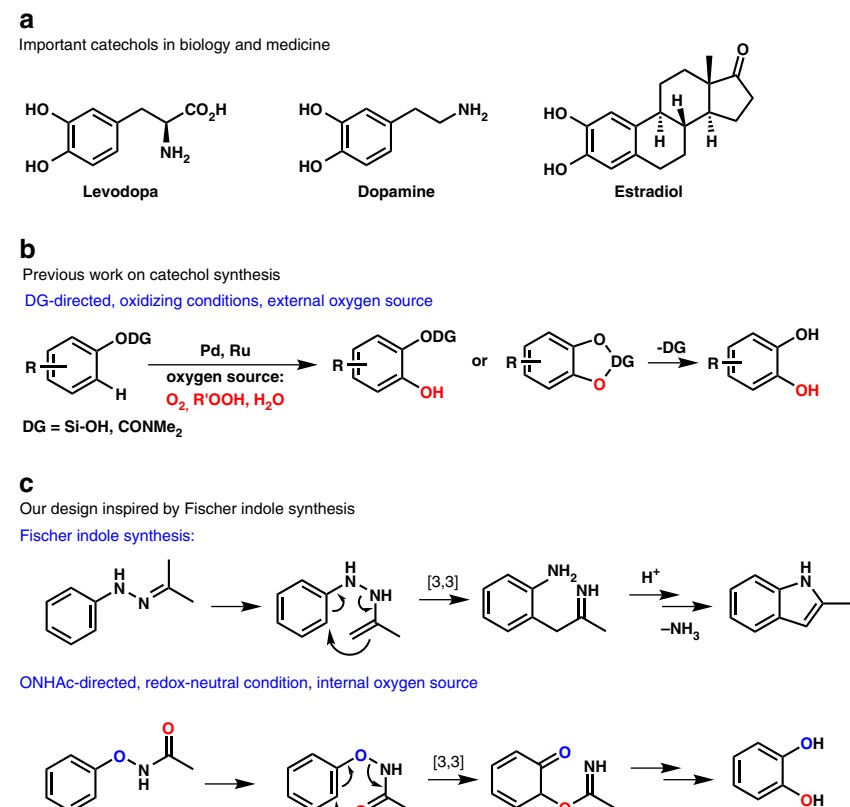

**Figure 1 | A new approach to catechol synthesis inspired by Fischer indole synthesis.** (**a**) Important catechols in biology and medicine. (**b**) Previous work on catechol synthesis. (**c**) Our design inspired by Fischer indole synthesis.

**Table 1 | Screening of reaction conditions.**

| Entry | Catalyst* | Additive | Solvent | O source | T (°C) | % yield† |
|---|---|---|---|---|---|---|
| 1 | ZnCl$_2$ | 2.5 eq. AcOH | Dioxane | — | 80/r.t. | ND |
| 2 | AlCl$_3$ | 2.5 eq. AcOH | Toluene | — | 80/r.t. | ND |
| 3 | FeCl$_3$ | 2.5 eq. AcOH | MeOH | — | 80/r.t. | ND |
| 4 | TsOH | — | MeOH | — | 80/r.t. | ND |
| 5 | Pd(OAc)$_2$ | 2 eq. KOAc/1 eq. BQ | DMA | 1 atm O$_2$ | 90 | ND |
| 6 | Pd(OAc)$_2$ | 2 eq. PhI(OAc)$_2$ | PhMe | PhI(OAc)$_2$ | 90 | ND |
| 7 | [Ru(p-cymene)Cl$_2$]$_2$ | 2 eq. K$_2$S$_2$O$_8$ | TFA/TFAA | TFA/TFAA | 80 | ND |
| 8 | Pd(OAc)$_2$ | 2 eq. K$_2$S$_2$O$_8$ | TFA/TFAA | TFA/TFAA | 80/r.t. | ND |
| 9 | Pd(OAc)$_2$ | 10 mol% PPh$_3$ | DCE | 1.2 eq. oxone | 90 | ND |
| 10 | [Cp*RhCl$_2$]$_2$ | — | MeOH | 2 eq. DTBP | 70 | ND |
| 11 | [Cp*IrCl$_2$]$_2$ | 2 eq. KOAc | MeOH | 2 eq. DTBP | 70 | ND |
| 12 | [Cp*IrCl$_2$]$_2$ | — | MeOH | 2 eq. DTBP | 70 | 25 |
| 13 | [Cp*IrCl$_2$]$_2$ | 2.5 eq. AcOH | MeOH | — | 70 | 30 |
| 14 | [Cp*IrCl$_2$]$_2$ | 2.5 eq. AcOH | MeOH | — | r.t. | 67 |
| 15 | [IrCl(COD)]$_2$ | 2.5 eq. AcOH | MeOH | — | 70/r.t. | ND |
| 16 | IrCl$_3$.H$_2$O | 2.5 eq. AcOH | MeOH | — | 70/r.t. | ND |
| 17 | [Ir(COD)OMe]$_2$ | 2.5 eq. AcOH | MeOH | — | 70/r.t. | ND |
| 18 | [Cp*IrCl$_2$]$_2$ | 2.5 eq. HCOOH | MeOH | — | r.t. | ND |
| 19 | [Cp*IrCl$_2$]$_2$ | 2.5 eq. CH$_2$(COOH)$_2$ | MeOH | — | r.t. | 92 |
| 20 | [Cp*IrCl$_2$]$_2$ | 2.5 eq. CH$_2$BrCOOH | MeOH | — | r.t. | 75 |
| 21 | [Cp*IrCl$_2$]$_2$ | 2.5 eq. TFA | MeOH | — | r.t. | ND |
| 22 | [Cp*IrCl$_2$]$_2$ | 2.5 eq. TFA/TFAA | MeOH | — | r.t. | ND |
| 23 | — | 2.5 eq. CH$_2$(COOH)$_2$ | MeOH | — | r.t. | ND |

ND, not detected; r.t., room temperature.
*Reaction conditions: 0.2 mmol **1a**, 5% mol catalyst, X mol additive in 0.1 M solvent under N$_2$.
†Isolated yield.

the design of the internal oxidation pathway, we made further attempts using other metal catalysts in the absence of an external oxidant. Gratifyingly, using 5 mol% [Cp*IrCl$_2$]$_2$ as catalyst and 2 equiv. of di-tert-butyl peroxide (DTBP) as a possible oxygen source and heating in MeOH at 70 °C for 12 h, the target product catechol **2a** was obtained in 25% yield (Table 1, entry 12). Concurrently, a mixture of phenol byproducts from the self-decomposition of N-phenoxyacetamides was detected. According to the literature and our past experience, the reaction temperature and acid–alkali environment are important factors in the self-decomposition of N-phenoxyacetamides. It was surprising to find that target product was synthesized in the absence of DTBP when one equivalent of AcOH was added (Table 1, entry 13). Lowering the temperature greatly improved the yield. At room temperature, the reaction proceeded to give 67% yield (Table 1, entry 14). Next, various acids and alkalis were tested to improve the yield. As a whole, acids were conducive to this reaction, whereas alkalis were not. Malonic acid was preferred, with a 95% gas chromatography (GC) yield and a 92% isolated yield (Table 1, entry 19). Bromoacetic acid was slightly inferior (Table 1, entry 20), whereas HCOOH, TFA and TFA/TFAA failed to afford the product (Table 1, entries 18, 21–22). When alkali was added, no product was observed (Table 1, entry 11). In the absence of [Cp*IrCl$_2$]$_2$, no product was detected (Table 1, entry 23). In addition, there was no distinction under N$_2$, O$_2$ and air as the reaction environment. Ultimately, optimal reaction conditions to carry the ortho-hydroxylation are: 5 mol% [Cp*IrCl$_2$]$_2$ and 2.5 equiv. of

CH$_2$(COOH)$_2$ in MeOH at room temperature in air with no external oxidant.

**Scope of Ir-catalysed ortho-hydroxylation.** The scope of the one-step neutral ortho-hydroxylation from N-phenoxyacetamides **1** to obtain catechols **2** was investigated (Table 2). Significantly, electron-neutral and electron-rich substituents such as Me, Et, tBu and OMe groups (**2a–2e**, **2o** and **2p**), all afforded excellent selectivity and good yields. The hydroxylation of substrates with electron-withdrawing substituents such as COOH, COOMe and halogen (F, Cl, Br and I with good yield) were also well tolerated (**2f–2n**). Ortho-, meta- and para-substituents on N-phenoxyacetamides provided similarly good yields. Remarkably, meta- and para-substituents on N-phenoxyacetamides shared the same products. This finding showed a good selectivity of hydroxylation in the ortho position, which was far away from the meta-substituents (**2c** and **2d**, **2j** and **2k**). In particular, the ortho-ethyl substituent substrate showed C–H activation on the aromatic ring rather than the aliphatic chain (**2e**).

**The synthesis of fluorescent and bioactive catechols.** Our method provided the possibility of one-step synthesis of complicated catechols (Table 3). Recently, we reported a new method to afford different fluorescent heteroarylated phenols by N-phenoxyacetamides[34]. When ortho-heteroarylated N-phenoxyacetamide **1u** was subject to the standard ortho-hydroxylation conditions, the desired product 2-(2,3-

**Table 2 | Scope of Ir-catalysed *ortho*-hydroxylation\*.**

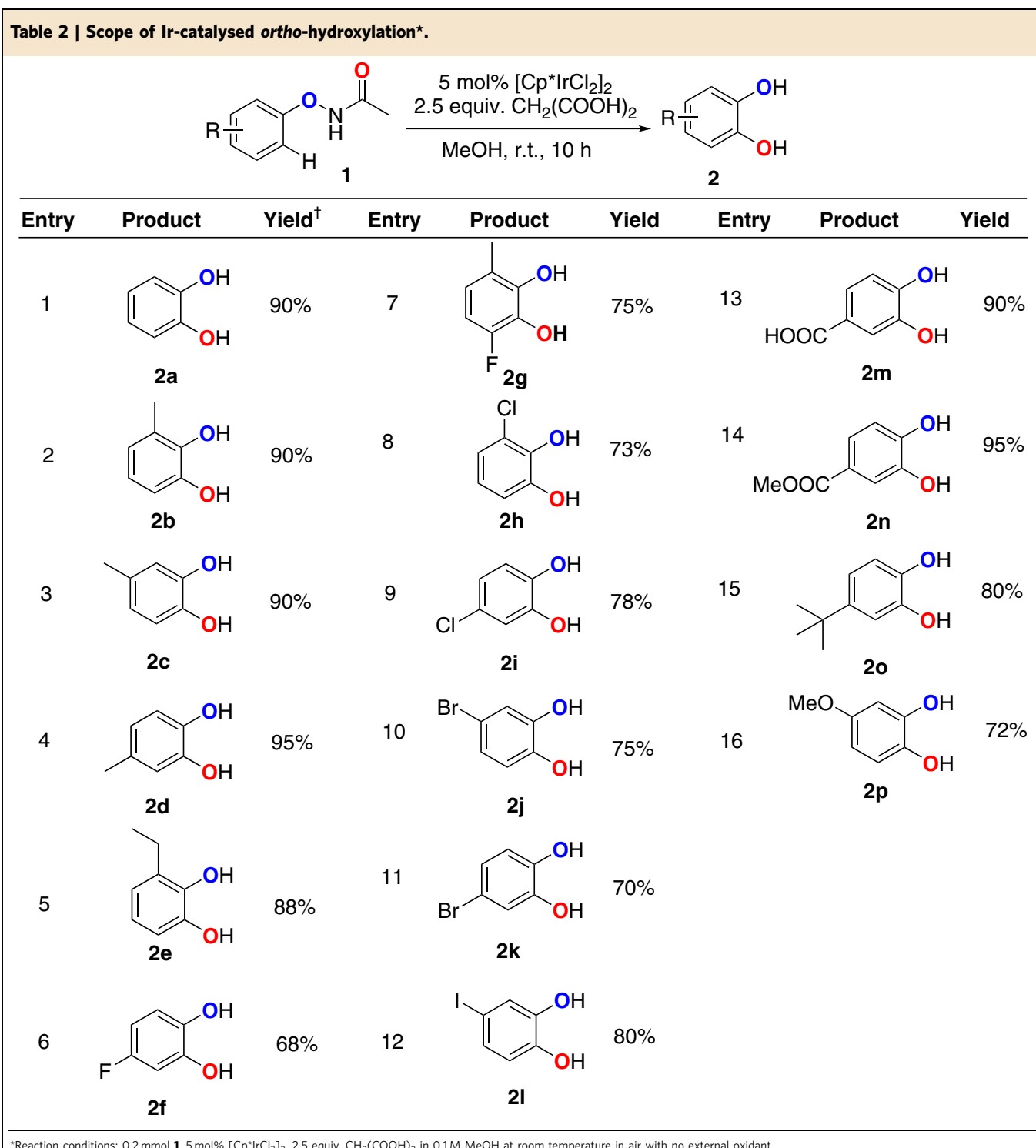

| Entry | Product | Yield[†] | Entry | Product | Yield | Entry | Product | Yield |
|---|---|---|---|---|---|---|---|---|
| 1 | 2a | 90% | 7 | 2g | 75% | 13 | 2m | 90% |
| 2 | 2b | 90% | 8 | 2h | 73% | 14 | 2n | 95% |
| 3 | 2c | 90% | 9 | 2i | 78% | 15 | 2o | 80% |
| 4 | 2d | 95% | 10 | 2j | 75% | 16 | 2p | 72% |
| 5 | 2e | 88% | 11 | 2k | 70% | | | |
| 6 | 2f | 68% | 12 | 2l | 80% | | | |

\*Reaction conditions: 0.2 mmol **1**, 5 mol% [Cp\*IrCl₂]₂, 2.5 equiv. CH₂(COOH)₂ in 0.1 M MeOH at room temperature in air with no external oxidant.
†Isolated yield.

dihydroxyphenyl)benzothiazole derivative **2u** with green fluorescence was obtained in 76% yield. Our method could be a general strategy to prepare fluorescent dihydroxyl heterocycle scaffold[48,49]. Next, we prepared the ONHAc-containing coumarins, the 7,8-dihydroxyl products **2v** and **2w**, with complete chemoselectivity in 87% and 81% yield, respectively. The product 7,8-dithydroxycoumarin **2v** was commonly called Daphnetin and is an important natural product from Zushima with good anti-inflammatory and anti-oxidant activities[50,51]. Our reaction could provide a powerful tool in the chemical synthesis

of Daphnetin derivatives. Finally, bioactive catechols natural products such as L-dopa and estradiol were tested under the standard reaction conditions, the protected L-dopa **2x**, protected dopamine **2y** and estradiol **2z** were successfully obtained directly with high selectivity and excellent yields, highlighting the method's mild conditions, excellent functional group tolerance and selectivity.

## Discussion

To understand the reaction mechanism, we first attempted to identify the oxygen source of the catechol products. With the

**Table 3 | Scope of fluorescent and bioactive catechols.**

| Entry | Substrate | | Product | | Yield[*,†] |
|-------|-----------|---|---------|---|-------|
| 1 |  | **1u** | <br>Fluorescent 2-DHPBT | **2u** | 76% |
| 2 |  | **1v** | <br>Daphentin | **2v** | 87% |
| 3 |  | **1w** | <br>Substituted Daphentin | **2w** | 81% |
| 4 |  | **1x** | <br>Protected L-dopa | **2x** | 80% |
| 5 |  | **1y** | <br>Protected Dopamine | **2y** | 85% |
| 6 |  | **1z** | <br>Estradiol | **2z** | 72% |

*Reaction conditions: 0.2 mmol **1**, 5 mol% [Cp*IrCl$_2$]$_2$, 2.5 equiv CH$_2$(COOH)$_2$ in 0.1 M MeOH at room temperature in air for 16 h.
†Isolated yield.

standard condition, using $^{18}$O-MeOH as the solvent (Fig. 2a, eq. 1), the product detected by high resolution mass spectroscopy (HRMS) did not show the $^{18}$O-labelled catechol, which meant the oxygen of the product did not come from the solvent. After that, 50 μl $^{18}$O-H$_2$O was added under the standard conditions (Fig. 2a, eq. 2). The results showed that the addition of water had no effect on the yield of catechol, but still no $^{18}$O-labelled product could be detected. It seemed as though that the oxygen of catechol did not come from the water that might be generated in the reaction. When $^{18}$O-labelled acetic acid (CH$_3$C$^{18}$O$^{18}$OH) was added instead of malonic acid (Fig. 2a, eq. 3), the $^{18}$O-labelled product was still not detected, which showed that the acid was also not the

**Figure 2 | Mechanistic study.** (**a**) Isotope labelling experiments. (**b**) Synthesis of [18]O-labelled catechols. (**c**) Seeking the active intermediate. (**d**) The importance of the N–H bond in the substrates.

oxygen source of the reaction. Based on the aforementioned results, we inferred that the oxygen of the product could only come from the carbonyl of the -O–NHAc group. Finally, we synthesized the substrate PhONHAc with the carbonyl labelled with [18]O. Under the standard conditions, we obtained the catechol with one [18]O-labelled hydroxyl (Fig. 2a, eq. 4). The [18]O-labelled substrate and product were confirmed by HRMS (Fig. 3), suggesting that the reaction was an intramolecular process and that the carbonyl of the -O–NHAc group was the oxygen source. This method may be a powerful tool to access a variety of [18]O-labelled catechols, which could then be used to investigate the metabolic pathways of catechol-derived bioactive compounds.

To probe the active intermediate in the reaction, we treated the substrate $N$-phenoxyacetamide (1 equiv.) and $[Cp^*IrCl_2]_2$ (0.5 equiv.) with $Ag_2CO_3$ (2 equiv.) in MeCN at room temperature for 10 h, and the five-membered iridium species **3** was obtained in 95% yield. The structure of complex **3** was confirmed by nuclear magnetic resonance spectroscopy, HRMS and X-ray crystallography. However, no product was obtained when 2.5 equiv. malonic acid was added in MeOH (Fig. 2c), implying that species **3** is not an active intermediate of the reaction. Furthermore, $N$-methyl-substituted phenoxyacetamide **1q** did not give the desired product (Fig. 2d, eq. 5) and most of the starting material was recovered. However, the reaction proceeded smoothly when the acetamide was replaced by hexanamide (Fig. 2d, eq. 6), highlighting the important role of the N–H bond.

We proposed that the seven-membered $[Ir^{III}]$ species **A** might be the active intermediate, which underwent the reductive

elimination process to form a C–O bond and afforded $[Ir^I]$ complex **B** (Fig. 4). Subsequently, the O–N bond oxidatively added to the iridium complex **B**, generating the $[Ir^{III}]$ complex **C**. Protonation of the intermediate **C** regenerated the original catalyst and the imine compound **D**, which would be hydrolysed to catechols.

The acid might play several roles such as preventing the generation of inactive intermediate **3**, consuming the byproduct ammonia and protonating the intermediate **C**. Further theoretical calculations revealed that when acid additives were introduced, the free energy of species **A** underwent a significant reduction from 22.4 to 2.1 kcal mol$^{-1}$, highlighting the crucial role of acids in stabling species **A** and protonation (Supplementary Fig. 1). Mass spectrometric experiments were carried out to explore the interaction between $[Cp^*IrCl_2]_2$ and different acids (HCOOH, $CH_2COOH$ and $CH_2(COOH)_2$) (Supplementary Fig. 2). Interestingly, in the Ir/HCOOH/MeOH system, HCOOH was detected losing a molecule $CO_2$ fragment when interacted with $[Cp^*IrCl_2]_2$, whereas $CH_2COOH$ or $CH_2(COOH)_2$ dissociated one molecule $CH_3COOH$ and $CH_2(COOH)_2$, respectively, by collision-induced dissociation. This result might explain why HCOOH was ineffective in promoting the reaction.

In summary, we have described the first examples of an iridium-catalysed synthesis of catechols from phenols through the formation of $N$-phenoxyacetamide intermediates. This bio-inspired route allows efficient, selective synthesis of catechols, which are useful building blocks for amino acids and pharmaceuticals. Our method also provides a convenient

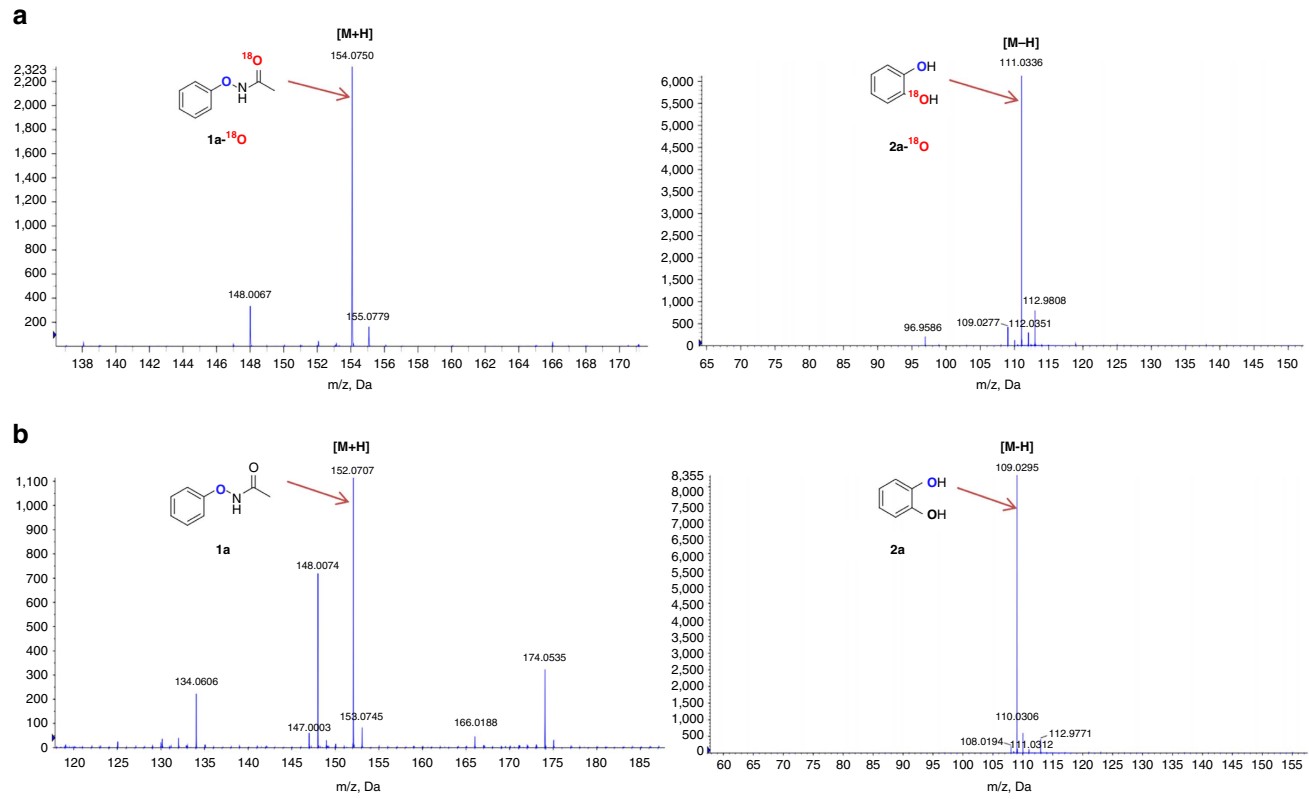

**Figure 3 | HRMS contrast of O[18]-labelled and non-O[18]-labelled substrate and product.** (**a**) HRMS of O[18]-labelled substrate and product. (**b**) HRMS of non-O[18]-labelled substrate and product.

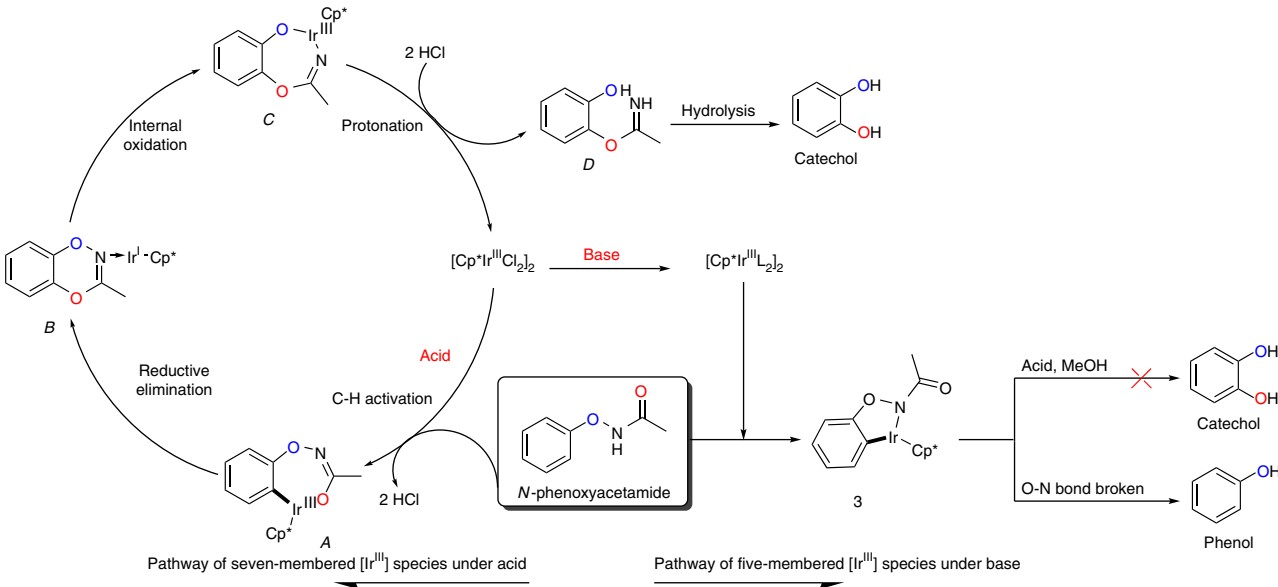

**Figure 4 | Proposed mechanism.** A plausible mechanism illustrating how catechols are formed in acidic condition (left) and phenols are formed in alkaline condition (right).

route to [18]O-labelled catechols using [18]O-labelled acetic acid. Further applications of the Ir/malonic acid/MeOH system and mechanistic studies of the reaction are under investigation and will be reported in due course.

## Methods

**Materials.** For NMR spectra of compounds in this study, see Supplementary Figs 1–27. For the crystallographic data of compound **3**, see Supplementary

Tables 1–2 and Supplementary Fig. 28. For the representative experimental procedures and analytic data of compounds synthesized, see Supplementary Methods.

**General procedure (2a).** N-phenoxyacetamide (**1**) (0.2 mmol), [Cp*IrCl$_2$]$_2$ (5 mol%) and CH$_2$(COOH)$_2$ (5.0 mmol) without external oxidant were weighed into a 10 ml pressure tube, to which was added anhydrous MeOH (1 ml) in a glove box. The reaction vessel was stirred at room temperature for 10 h in air. The mixture was then concentrated under vacuum and the residue was purified by

column chromatography on silica gel with a gradient eluent of petroleum ether and ethyl acetate to afford the corresponding product.

**Data availability.** The X-ray crystallographic coordinates for structures reported in this study have been deposited at the Cambridge Crystallographic Data Centre, under deposition number CCDC14911482. These data can be obtained free of charge from The Cambridge Crystallographic Data Centre via www.ccdc.cam.ac. uk/data_request/cif. The authors declare that all other data supporting the findings of this study are available within the article and Supplementary Information files, and also are available from the corresponding author upon reasonable request.

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

## Acknowledgements

Financial support was provided by the Shenzhen Government (JCYJ20150626110741712 and GJHZ20140417144118026), the Guangdong Government (S20120011226), the National Science Foundation of China (21622103 and 21571098) and the MOST of China (2014AA020512).

## Author contributions

Y.C. performed most of the reaction optimization. The experimental work was conducted by Y.C., D.Y.Y. and Q.W. Q.W. prepared most of the manuscript and

supporting information. J.Z. conceived the initial ideas of this work, supervised all the experiments and coordinated with Y.L. and W.Y.S.

## Additional information

**Competing financial interests:** The authors declare no competing financial interests.

**Publisher's note**: 

