## [Peer review file · Nature Communications]

Reviewers' comments:

Reviewer #1 (Remarks to the Author):

Selective oxidation is a key reaction in organic synthesis. In this manuscript, the authors developed an Ir-catalyzed tyrosinase-like approach to catechols via the formation of N-phenoxyacetamide intermediates, which are very useful building blocks for organic synthesis and pharmaceuticals. The experiment of the isotope labeling supported the plausible reaction mechanism. Moreover, this protocol provided an alternative strategy to oxyacetamide-directed C–H hydroxylation on phenols and achieves one-pot synthesis of catechols with diverse substituent groups in good to excellent yield and high selectivity under mild conditions. Especially the examples 2x, 2y, and 2z are very interesting and useful substrates. In my opinion, all the synthesis strategy represented the atom and step economical of direct C-H functionalization in organic chemistry, this report deserves to be published in the Nature Commun.

Reviewer #2 (Remarks to the Author):

The authors described the development of a new iridium-catalyzed tyrosinase-like approach to catechols, which used an oxyacetamide-directed C–H hydroxylation on phenols. This method can access catechols quickly via one-step, redox-neutral synthesis protocol and a variety of substituted catechol derivatives were prepared under mild conditions. Further mechanistic studies revealed that the directing group (DG) oxyacetamide served as the oxygen source in this reaction. The authors also prepared ^{18}O -labeled catechols by using ^{18}O -labeled acetic acid. The authors further demonstrated that this synthetic strategy can be applied to the synthesis of dopamine, L-dopa and estradiol from the corresponding phenols.

Overall, I think this is a quite interesting and useful work for providing a new protocol to synthesize catechols, which can be biologically prepared by ubiquitous tyrosinase catalyzes the aerobic oxidation of phenols to through the binuclear copper centers. I would like to recommend acceptance for publication in NC after addressing following questions.

Question 1:

The reaction scope need further investigations. For example, if heterocycle will be compatible in this reaction.

Question 2:

The authors should mention the easiness of starting materials.

Question 3:

If authors can do some computational study to reveal why Ir is capable is this re-dox reaction ? How about different Rh catalysts and if authors had extensively tried Rh catalysts ?

Question 4:

If authors can explain the role of acid additives, which seems to be a key thing for this new reaction.

Detailed Responses to Referee's Comments

Reviewers' comments:

Reviewer #1 (Remarks to the Author):

Selective oxidation is a key reaction in organic synthesis. In this manuscript, the authors developed an Ir-catalyzed tyrosinase-like approach to catechols via the formation of N-phenoxyacetamide intermediates, which are very useful building blocks for organic synthesis and pharmaceuticals. The experiment of the isotope labeling supported the plausible reaction mechanism. Moreover, this protocol provided an alternatively strategy to oxyacetamide-directed C–H hydroxylation on phenols and achieves one-pot synthesis of catechols with diverse substituent groups in good to excellent yield and high selective under mild conditions. Especially the example 2x, 2y, and 2z are very interesting and useful substrates. In my opinion, all the synthesis strategy represented the atom and step economical of direct C-H functionalization in organic chemistry, this report deserves to be published in the Nature Commun.

Response

We thank the reviewer for this nice comment.

Reviewer #2 (Remarks to the Author):

The authors described the development of a new iridium-catalyzed tyrosinase-like approach to catechols, which used an oxyacetamide-directed C–H hydroxylation on phenols. This method can access catechols quickly via one-step, redox-neutral synthesis protocol and a variety of substituted catechol derivatives were prepared under mild conditions. Further mechanistic studies revealed that the directing group (DG) oxyacetamide served as the oxygen source in this reaction. The authors also prepared ¹⁸O-labeled catechols by using ¹⁸O-labeled acetic acid. The authors further demonstrated that this synthetic strategy can be applied to the synthesis of dopamine, L-dopa and estradiol from the corresponding phenols.

Overall, I think this is a quite interesting and useful work for providing a new protocol to synthesize catechols, which can be biologically prepared by ubiquitous tyrosinase catalyzes the aerobic oxidation of phenols to through the binuclear copper centers. I would like to recommend acceptance for publication in NC after addressing following questions.

Question 1:

The reaction scope need further investigations. For example, if heterocycle will be compatible in this reaction.

Response

We thank the reviewer for this valuable advice. We made further investigations on one-step synthesis of complicated catechols with fluorescent property and bioactivity. Recently we reported a new method to afford different fluorescent heteroarylated

phenols by *N*-phenoxyacetamides. When *ortho*-heteroarylated *N*-phenoxyacetamide **1u** was subject to the standard *ortho*-hydroxylation conditions, the desired product 2-(2,3-dihydroxyphenyl)benzothiazole (2-DHPBT) derivative **2u** with green fluorescence was obtained in 76% yield. Our method could be a general strategy to prepare fluorescent dihydroxyl heterocycle scaffold. Next, we prepared the ONHAc-containing coumarins, the 7,8-dihydroxyl products **2v** and **2w**, with complete chemoselectivity in 87% and 81% yield respectively. The product 7,8-dihydroxycoumarin **2v** was commonly called Daphnetin and is an important natural product from Zushima with good anti-inflammatory and anti-oxidant activities. Our reaction could provide a powerful tool in the chemical synthesis of Daphnetin derivatives.

Revisions Made

(Please refer to page 3, paragraph 3, Table 3 and supporting information).

One-pot synthesis of different catechols with fluorescent property and bioactivity. Our method provided the possibility of one-step synthesis of complicated catechols (**Table 3**). Recently we reported a new method to afford different fluorescent heteroarylated phenols by *N*-phenoxyacetamides.³⁴ When *ortho*-heteroarylated *N*-phenoxyacetamide **1u** was subject to the standard *ortho*-hydroxylation conditions, the desired product 2-(2,3-dihydroxyphenyl) benzothiazole (2-DHPBT) derivative **2u** with green fluorescence was obtained in 76% yield. Our method could be a general strategy to prepare fluorescent dihydroxyl heterocycle scaffold.^{48,49} Next, we prepared the ONHAc-containing coumarins, the 7,8-dihydroxyl products **2v** and **2w**, with complete chemoselectivity in 87% and 81% yield respectively. The product 7,8-dihydroxycoumarin **2v** was commonly called Daphnetin and is an important natural product from Zushima with good anti-inflammatory and anti-oxidant activities.^{50,51} Our reaction could provide a powerful tool in the chemical synthesis of Daphnetin derivatives. Finally, bioactive catechols natural products such as L-dopa and estradiol were tested under the standard reaction conditions, the protected L-dopa **2x**, protected dopamine **2y** and estradiol **2z** were successfully obtained directly with high selectivity and excellent yields, highlighting the method's mild conditions, excellent functional group tolerance, and selectivity.

34. Wu, Q., Chen, Y., Yan, D., Zhang, M., Lu, Y., Sun, W. Y., & Zhao, J. Unified synthesis of mono/bis-arylated phenols via Rh^{III}-catalyzed dehydrogenative coupling. *Chem. Sci.* Advance Article (2016).

48. Charles, R. G. & Freiser, H. Synthesis of 2-(*O*-hydroxyphenyl) benzothiazole and of 2-(*O*-hydroxyphenyl) benzothiazoline. *J. Org. Chem.* **18**, 422-425 (1953).

49. Shindo, K., Shindo, Y., Hasegawa, T., Osawa, A., Kagami, O., Furukawa, K., & Misawa, N. Synthesis of highly hydroxylated aromatics by evolved biphenyl dioxygenase and subsequent dihydrodiol dehydrogenase. *Appl. Microbiol. Biot.* **75**, 1063-1069 (2007).

50. Molyneux, R. J., & Jurd, L. E. O. N. A. R. D. The condensation of some phenols with malic acid, maleic acid and maleic

anhydride. *Aust. J. Chem.* **27**, 2697-2702 (1974).

51. Fylaktakidou, K. C., Hadjipavlou-Litina, D. J., Litinas, K. E., & Nicolaidis, D. N. Natural and synthetic coumarin derivatives with anti-inflammatory/antioxidant activities. *Curr. Pharm. Design* **10**, 3813-3833(2004).

Table 3 | Scope of fluorescent and bioactive catechols

Entry	Substrate	Product	Yield ^{a,b}
1		 Fluorescent 2-DHPBT	76%
2		 Daphentin	87%
3		 Substituted Daphentin	81%
4		 Protected L-dopa	80%
5		 Protected Dopamine	85%
6		 Estradiol	72%

^a Reaction conditions: 0.2 mmol **1**, 5 mol% [Cp*IrCl₂]₂, 2.5 equiv CH₂(COOH)₂ in 0.1 M MeOH at room temperature in air for 16 h. ^b Isolated yield.

Question 2:

The authors should mention the easiness of starting materials.

Response

We thank the reviewer for this kind suggestion. A variety of substituted substrates with O–NHAc directing group were easily prepared from phenols or aryl boronic acid compounds. Detailed synthesis procedures had been displayed in supporting information.

Revisions Made

(Please refer to page 2, paragraph 2 and supporting information II).

A variety of substituted substrates with -O–NHAc directing group were easily prepared from phenols or aryl boronic acid compounds.^{35-37, 41}

35. Endo, Y., Shudo, K., & Okamoto, T. An acid catalyzed rearrangement of *O*-aryl-*N*-benzoylhydroxylamines; synthesis of catechols from phenols. *Synthesis*, **6**, 461-463 (1980).
36. Li, B., Lan, J., Wu, D., & You, J. Rhodium(III)-catalyzed *ortho*-heteroarylation of phenols through Internal oxidative C-H activation: rapid screening of single-molecular white-light-emitting materials. *Angew. Chem. Int. Ed.* **127**, 14214-14218 (2015).
37. Petrassi, H. M., Sharpless, K. B., & Kelly, J. W. The copper-mediated cross-coupling of phenylboronic acids and *N*-hydroxyphthalimide at room temperature: synthesis of aryloxyamines. *Org. Lett.* **3**, 139-142 (2001).
41. Liu, G., Shen, Y., Zhou, Z. & Lu, X. Rhodium (III)-catalyzed redox-neutral coupling of *N*-phenoxyacetamides and alkynes with tunable selectivity. *Angew. Chem. Int. Ed.* **52**, 6033-6037 (2013).

Question 3:

If authors can do some computational study to reveal why Ir is capable is this re-dox reaction? How about different Rh catalysts and if authors had extensively tried Rh catalysts?

Response

We thank the reviewer for this helpful suggestion. Different kinds of catalysts had been screened and revealed in Supplementary Table 1. Rh catalyst with different valence state didn't work on this reaction. Regarding to the difference between Ir and Rh, we think that the size and redox behaviour of the two metals might contribute to the fine-tuning of their catalytic activity.

Revisions Made

The results were showed in revised manuscript.

(Please refer to page 2, paragraph 3 and supporting information III).

We first explored Zn-, Al-, Fe-, Cu-, Pd- and Ru-catalyzed systems to *N*-phenoxyacetamides (**1a**). Unfortunately, the target product (**2a**) was not detected. The same result was shown in the common Rh-catalyzed system for -O–NHAc (**Table 1, entries 1–10** and **Supplementary Table 1**).

Table S1. The screen of reaction conditions.

entry	catalyst ^a	additive	solvent	O source	T/°C	% yield ^b
1	[(COD)Rh(acac)] ₂	2.5 eq. CH ₂ (COOH) ₂	MeOH	—	r.t.	ND
2	[(COD)RhCl] ₂	2.5 eq. CH ₂ (COOH) ₂	MeOH	—	r.t.	ND
3	Rh(OAc) ₂	2.5 eq. CH ₂ (COOH) ₂	MeOH	—	r.t.	ND
4	[Cp*RhCl ₂] ₂	2.5 eq. CH ₂ (COOH) ₂	MeOH	—	r.t.	ND
5	Pd(OAc) ₂	2.5 eq. CH ₂ (COOH) ₂	MeOH	—	r.t.	ND
6	Pb(acac) ₂	2.5 eq. CH ₂ (COOH) ₂	MeOH	—	r.t.	ND
7	Pb(PPh ₃) ₄	2.5 eq. CH ₂ (COOH) ₂	MeOH	—	r.t.	ND
8	Pb(TFA) ₂	2.5 eq. CH ₂ (COOH) ₂	MeOH	—	r.t.	ND
9	[Cp*IrCl ₂] ₂	2.5 eq. HOOC-COOH	MeOH	—	r.t.	ND
10	[Cp*IrCl ₂] ₂	2.5 eq. TsOH	MeOH	—	r.t.	ND
11	[Cp*IrCl ₂] ₂	2.5 eq. CH ₂ CHCOOH	MeOH	—	r.t.	ND
12	[Cp*IrCl ₂] ₂	2.5 eq. TMBA	MeOH	—	r.t.	63
13	[Cp*IrCl ₂] ₂	2.5 eq. p -Toluic acid	MeOH	—	r.t.	46
14	[Cp*IrCl ₂] ₂	2.5 eq. 1-adamantoic acid	MeOH	—	r.t.	54
15	[Cp*IrCl ₂] ₂	2.5 eq. CH ₂ (COOH) ₂	MeOH	—	r.t.	92
16	[Cp*IrCl ₂] ₂	2.5 eq. CH ₂ (COOH) ₂	EtOH	—	r.t.	73
17	[Cp*IrCl ₂] ₂	2.5 eq. CH ₂ (COOH) ₂	ⁱ PrOH	—	r.t.	23
18	[Cp*IrCl ₂] ₂	2.5 eq. CH ₂ (COOH) ₂	AmOH	—	r.t.	ND
19	[Cp*IrCl ₂] ₂	2.5 eq. CH ₂ (COOH) ₂	Ethylene glycol	—	r.t.	15
20	[Cp*IrCl ₂] ₂	2.5 eq. CH ₂ (COOH) ₂	MeCN	—	r.t.	ND
21	[Cp*IrCl ₂] ₂	2.5 eq. CH ₂ (COOH) ₂	DMSO	—	r.t.	ND
22	[Cp*IrCl ₂] ₂	2.5 eq. CH ₂ (COOH) ₂	THF	—	r.t.	ND

Reaction conditions: 0.2 mmol **1a**, 5 mol% catalyst, 5.0 mmol acid in 0.1 M solvent at room temperature under N₂ for 12 h. ^b GC yield.

Question 4:

If authors can explain the role of acid additives, which seems to be a key thing for this new reaction.

Response

We thank the reviewer for this helpful suggestion. Following this suggestion, we did two part computational study for evaluating the significance of acid additives. Theoretical calculations revealed that when acid additives were introduced, the free energy of species **A** underwent a significant reduction from 22.4 kcal/mol to 2.1 kcal/mol, highlighting the crucial role of acids in stabilizing species **A** and protonation. Further mass spectrometric experiments were carried out to explore the interaction between [Cp*IrCl₂]₂ and different acids (HCOOH, CH₂COOH and CH₂(COOH)₂). Interestingly, in the Ir/HCOOH/MeOH system, HCOOH was detected losing a molecule CO₂ fragment when interacted with [Cp*IrCl₂]₂ while CH₂COOH or CH₂(COOH)₂ dissociated one molecule CH₃COOH and CH₂(COOH)₂ respectively by collision induced dissociation (CID). This result might explain why HCOOH was ineffective in promoting the reaction.

Revisions Made

(Please refer to page 5, paragraph 2 and supporting information V)

The acid might play several roles, such as preventing the generation of inactive intermediate **3**, consuming the by-product ammonia, and protonating the intermediate **C**. Further theoretical calculations revealed that when acid additives were introduced, the free energy of species **A** underwent a significant reduction from 22.4 kcal/mol to 2.1 kcal/mol, highlighting the crucial role of acids in stabilizing species **A** and protonation (**Supplementary Fig. S1**). Mass spectrometric experiments were carried out to explore the interaction between $[\text{Cp}^*\text{IrCl}_2]_2$ and different acids (HCOOH , CH_2COOH and $\text{CH}_2(\text{COOH})_2$) (**Supplementary Fig. S2**). Interestingly, in the $\text{Ir}/\text{HCOOH}/\text{MeOH}$ system, HCOOH was detected losing a molecule CO_2 fragment when interacted with $[\text{Cp}^*\text{IrCl}_2]_2$ while CH_2COOH or $\text{CH}_2(\text{COOH})_2$ dissociated one molecule CH_3COOH and $\text{CH}_2(\text{COOH})_2$ respectively by collision induced dissociation (CID). This result might explain why HCOOH was ineffective in promoting the reaction.

Supporting information:

V. Mechanism study

b) Theoretical calculations

Computational details:

All the calculations were performed with Gaussian 09 package.⁵ Geometry optimization of all stationary points in gas phase was conducted by B3LYP method.⁶ The LANL2DZ basis set⁷ with ECP was used for Ir, and the 6-31G(d) basis set⁸ was used for other atoms. Frequency analysis was performed for each optimized structure to verify the stationary points as either minima or saddle point at the same level of theory. Solvent effect (solvent=methanol) was included by single-point energy calculation using SMD⁹ solvation model and B3LYP-D3 method¹⁰ with the def2-TZVP basis set for Ir and 6-311++G(d, p) basis set for the other atoms.¹¹ All relative energies (corrected with zero point energy) and Gibbs free energies (at 298.15 K and 1 atm) are reported in kcal mol^{-1} .

Figure S1 Intermediate **A** and intermediate **3** under natural state (a) and protonated state (b). Relative free energy and electronic energy (in parentheses) in solution are given in kcal/mol.

For natural mode, the free energy of species **A** is about 22.4 kcal/mol much higher than that of intermediate **3**, which indicates that in natural or basic condition intermediate **3** is favored.

While by introducing acid additives, species **A** could be stabilized to a large extent. The energy difference is only about 2.1 kcal/mol between protonated **A-H⁺** and the **3-H⁺** one. This two species are compatible.

c) Mass spectrometric experiments

Experimental details:

Experiments were carried out using a Synapt G2-S high definition mass spectrometer with an electrospray ion source. The instrument was held at a capillary voltage of 3.0 kV. The instrument has a quadrupole mass filter for the selection of parent ions. Mass selected ions enter into a linear ion trap cell which is filled with argon. The pressure is approximately 8×10^{-3} mbar. After collision, a reflectron time-of-flight (TOF) detector was used to determine the masses. The ion source block and nitrogen desolvation gas temperatures were set to 100 °C and 300 °C. For a given ion of interest, the elemental composition was confirmed by examination of the associated isotope envelopes in the source spectra. Collision induced dissociation was employed to analyze structural fragments.

The samples were diluted to millimolar by methanol (acid 0.37mM, $[\text{Cp}^*\text{IrCl}_2]_2$ 0.37mM), and then infused into ESI source at a flow rate of $5 \mu\text{l min}^{-1}$ through a syringe pump.

Figure S2 (a) CID mass spectrum of ion $m/z=737.15$ at $U_{\text{trap}}=10$ V ($E=0.52$ eV) in 1:1 mixture of $[\text{Cp}^*\text{IrCl}_2]_2$ with HCOOH solution. (b) CID mass spectrum of ion $m/z=751.17$ at $U_{\text{trap}}=12$ V ($E=0.61$ eV) in 1:1 mixture of $[\text{Cp}^*\text{IrCl}_2]_2$ with CH₃COOH solution. (c) CID mass spectrum of ion $m/z=795.15$ at $U_{\text{trap}}=17$ V ($E=0.81$ eV) in 1:1 mixture of $[\text{Cp}^*\text{IrCl}_2]_2$ with CH₂(COOH)₂ solution. (Ar is used as collision gas.)

A series of methanol solutions of a 1:1 mixture of $[\text{Cp}^*\text{IrCl}_2]_2$ (0.37 mM) with HCOOH, CH₂COOH, CH₂(COOH)₂ acids were subjected to ESI-MS analysis respectively. Peaks $m/z=737$, 751 and 795 were observed in the three different samples, were preliminarily assigned to $[\text{Cp}^*_2\text{Ir}_2\text{ClCHO}_2]^+$, $[\text{Cp}^*_2\text{Ir}_2\text{ClC}_2\text{H}_4\text{O}_2]^+$, $[\text{Cp}^*_2\text{Ir}_2\text{ClC}_3\text{H}_4\text{O}_4]^+$. To further characterize the structure of these ions, collision induced dissociation (CID) were performed. After collision with Ar, ions $m/z=751$ and 795 in CH₃COOH, CH₂(COOH)₂ solutions dissociate one molecule CH₃COOH and CH₂(COOH)₂ respectively (Figure S2(b) and (c)). However, the ion $m/z=737$ in

HCOOH system, loses a molecule CO₂ fragment (Figure S2(a)). This may give a hint why HCOOH failed for this reaction while with the addition of CH₃COOH or CH₂(COOH)₂ can afford the target product. The preliminary mass spectrometric result shows different reaction patterns in the presence of different acids, which may relate to the role of acid. Further comprehensive mechanistic studies are undergoing and will be covered in a following full paper.

- (5) Frisch, M. J., Trucks, G. W., Schlegel, H. B., Scuseria, G. E., Robb, M. A., Cheeseman, J. R., Scalmani, G., Barone, V., Mennucci, B., Petersson, G. A., Nakatsuji, H., Caricato, M., Li, X., Hratchian, H. P., Izmaylov, A. F., Bloino, J., Zheng, G., Sonnenberg, J. L., Hada, M., Ehara, M., Toyota, K., Fukuda, R., Hasegawa, J., Ishida, M., Nakajima, T., Honda, Y., Kitao, O., Nakai, H., Vreven, T., Montgomery, J. A., Jr., Peralta, J. E., Ogliaro, F., Bearpark, M., Heyd, J. J., Brothers, E., Kudin, K. N., Staroverov, V. N., Keith, T., Kobayashi, R., Normand, J., Raghavachari, K., Rendell, A., Burant, J. C., Iyengar, S. S., Tomasi, J., Cossi, M., Rega, N., Millam, J. M., Klene, M., Knox, J. E., Cross, J. B., Bakken, V., Adamo, C., Jaramillo, J., Gomperts, R., Stratmann, R. E., Yazyev, O., Austin, A. J., Cammi, R., Pomelli, C., Ochterski, J. W., Martin, R. L., Morokuma, K., Zakrzewski, V. G., Voth, G. A., Salvador, P., Dannenberg, J. J., Dapprich, S., Daniels, A. D., Farkas, O., Foresman, J. B., Ortiz, J. V., Cioslowski, J. & Fox, D. J. *Gaussian 09, Rev. D.01*, Gaussian, Inc.: Wallingford, CT, 2010.
- (6) a) Becke, A. D. Density-functional exchange-energy approximation with correct asymptotic behavior. *Phys. Rev. A: Gen. Phys.* **38**, 3098-3100 (1988); b) Becke, A. D. Density-functional thermochemistry. III. The role of exact exchange. *J. Chem. Phys.* **98**, 5648-5652 (1993); c) Lee, C., Yang, W. & Parr, R. G. Development of the Colle-Salvetti correlation-energy formula into a functional of the electron density. *Phys. Rev., B* **37**, 785-789 (1988).
- (7) a) Hay, P. J. & Wadt, W. R. Ab initio effective core potentials for molecular calculations. Potentials for K to Au including the outermost core orbitals. *J. Chem. Phys.* **82**, 299-310 (1985); b) Ehlers, A. W., Böhme, M., Dapprich, S., Gobbi, A., Höllwarth, A., Jonas, V., Köhler, K. F., Stegmann, R., Veldkamp, A. & Frenking, G. A set of f-polarization functions for pseudo-potential basis sets of the transition metals Sc-Cu, Y-Ag and La-Au. *Chem. Phys. Lett.* **208**, 111-114 (1993); c) Roy, L. E., Hay, P. J. & Martin, R. L. Revised Basis Sets for the LANL Effective Core Potentials. *J. Chem. Theory Comput.* **4**, 1029-1031 (2008).
- (8) a) Ditchfield, R., Hehre, W. J. & Pople, J. A. Self-consistent molecular-orbital methods. IX. An extended gaussian-type basis for molecular-orbital studies of organic molecules. *J. Chem. Phys.* **54**, 724-728 (1971); b) Hariharan, P. C. & Pople, J. A. The influence of polarization functions on molecular orbital hydrogenation energies. *Theor. Chim. Acta.* **28**, 213-222 (1973).
- (9) Marenich, A. V., Cramer, C. J. & Truhlar, D. G. Universal solvation model based on solute electron density and on a continuum model of the solvent defined by the bulk dielectric constant and atomic surface tensions. *J. Phys. Chem. B* **113**, 6378-6396 (2009).
- (10) Grimme, S., Antony, J., Ehrlich, S. & Krieg, H. A consistent and accurate ab initio parametrization of density functional dispersion correction (DFT-D) for the 94 elements H-Pu. *J. Chem. Phys.* **132**, 154104 (2010).
- (11) a) Weigend, F. & Ahlrichs, R. Balanced basis sets of split valence, triple zeta valence and quadruple zeta valence quality for H to Rn: Design and assessment of accuracy. *Phys. Chem. Chem. Phys.* **7**, 3297-3305 (2005); b) Krishnan, R., Binkley, J. S., Seeger, R. & Pople, J. A. Self-consistent molecular orbital methods. XX. A basis set for correlated wave functions. *J. Chem. Phys.* **72**, 650-654 (1980); c) Clark, T., Chandrasekhar, J., Spitznagel, G. W. & Schleyer, P. V. R. Efficient diffuse function-augmented basis sets for anion calculations. III. The 3-21+G basis set for first-row elements, Li-F. *J. Comp. Chem.* **4**, 294-301 (1983); d) The basis sets were obtained from the Gaussian Basis Set Library EMSL at <https://bse.pnl.gov/bse/portal>.

Reviewer #2 (Remarks to the Author):

After reviewing the revised manuscripts, I am satisfied with all revisions made by authors. I will recommend the acceptance of this article in its current form.